# Genetic Diversity and New Sequence Types of *Escherichia coli* Coharboring β-Lactamases and PMQR Genes Isolated from Domestic Dogs in Central Panama

**DOI:** 10.3390/genes14010073

**Published:** 2022-12-26

**Authors:** Virginia Núñez-Samudio, Gumercindo Pimentel-Peralta, Alexis De La Cruz, Iván Landires

**Affiliations:** 1Instituto de Ciencias Médicas, Las Tablas 0710, Los Santos, Panama; 2Sección de Epidemiología, Departamento de Salud Pública, Región de Salud de Herrera, Ministry of Health, Chitré 0601, Herrera, Panama; 3Escuela de Biología, Facultad de Ciencias Naturales, Exactas y Tecnología, Centro Regional Universitario de Azuero (CRUA), Universidad de Panamá, Chitré 0601, Herrera, Panama; 4Water Quality Laboratory, Ministry of Health, La Villa 0739, Los Santos, Panama; 5Hospital Joaquín Pablo Franco Sayas, Región de Salud de Los Santos, Ministry of Health, Las Tablas 0710, Los Santos, Panama

**Keywords:** extended-spectrum β-lactamase, canines, multilocus sequence typing, *Escherichia coli*, antimicrobial resistance, Panama

## Abstract

Background: β-lactamase-producing *Escherichia coli* are a widely distributed source of antimicrobial resistance for animals and humans. Little is known about the susceptibility profile and genetic characteristics of *E. coli* strains isolated from domestic dogs in Latin America. Methods: We report on a cross-sectional study that evaluated *E. coli* strains isolated from fecal samples of domestic dogs in central Panama. The extended-spectrum β-lactamase (ESBL), AmpC genes, and plasmid-mediated quinolone resistance were investigated. Molecular typing using Pasteur’s multilocus sequence typing (MLST) was conducted. Results: A total of 40 *E. coli* isolates were obtained, of which 80% (32/40) were resistant to at least one of the antibiotics tested, while 20% (8/40) were sensitive to all antibiotics analyzed in this study (*p* < 0.001). Forty percent of the strains were resistant to three or more antibiotics. The most common resistance was to tetracycline (45%) and ampicillin (30%) while 2.5% showed an ESBL phenotype. Antibiotic resistance genes were detected for one β-lactamase (blaTEM-1) and two plasmid-mediated quinolone resistance (PMQR) enzymes (qnrS and qnrB). In addition, mutations in the chromosomal *Amp*C gene were observed at positions −35, −28, −18, −1, and +58. Fourteen different sequence types (STs) were identified; the most frequent were ST399 and ST425 (12% each). ST3 and ST88, which have been previously identified in human clinical isolates, were also evidenced. Three new STs were found for the first time: ST1015, ST1016 (carrier of the *bla*TEM-1 gene), and ST1017 (carrier of the *bla*TEM-1, *qnrS*, and *qnrB* genes). Conclusions: In the intestinal strains of *E. coli* isolated from domestic dogs, there was a high frequency of resistance to antibiotics. The presence of genes from plasmids and chromosomal mutations that conferred antibiotic resistance, the identification of isolates previously reported in humans, and the genetic diversity of STs (including three that were newly identified) confirmed the determinants of resistance to antibiotics in the domestic dogs from central Panama.

## 1. Introduction

*Escherichia coli* is a commensal bacterium of the intestinal tract. While some strains of *E. coli* can cause intestinal infections, most strains can potentially cause extra-intestinal infections in many cases resistant to antibiotics, especially in the urinary tract. Therefore, under conditions favoring pathogenicity, *E. coli* is responsible for many infections in both humans and in animals [1]. These infections are mainly treated with β-lactam and fluoroquinolone antibiotics. However, resistance to these antibiotic groups has increased worldwide, which represents a major public health problem [2,3].

A study conducted in Latin America reported that the antibiotics most used for the treatment of companion animal infections were critically important and included β-lactams and quinolones, which were prescribed in 65.3% and 36.2% of infections, respectively [4]. Indiscriminate use of antibiotics in both human and veterinary medicine has resulted in bacterial resistance to antibiotics used in humans and their companion animals [5]. The exchange of bacterial serotypes between humans and domestic dogs has been reported [6,7,8,9,10,11].

Extended-spectrum β-lactamase-producing (ESBL) Enterobacteriaceae are a widely distributed source of antimicrobial resistance (AMR) in both humans and animals [12]. ESBL enzymes confer resistance to third- and fourth-generation cephalosporins and aztreonam. According to a systematic review conducted in South America, the prevalence of animal-origin ESBL-producing *E. coli* was estimated at 18.1% [13]. In addition, the prevalence of human-origin ESBL-producing *E. coli* in South America was estimated at 30% [14]. These results were similar to those observed in Tanzania, where ESBLs from animal sources showed a prevalence comparable to that of human isolates [13]. Thus, although many of the resistance mechanisms of *E. coli* are acquired in the human hospital environment, in several regions of the world the prevalence of ESBL-producing *E. coli* in animal sources can be considered a major problem. The latter may be due to the poor regulation of antibiotic use in these settings.

ESBL enzymes have been commonly detected in *E. coli* strains from healthy companion animals; the *bla*CTX-M-type, *bla*SHV, and *bla*TEM genes are the most identified on all continents [15]. The prevalence of the ESBL phenotype detected in *E. coli* isolates from dogs varies by region. For example, prevalence-rate estimates of 6% in Mexico [16], 9% in New Zealand [17], 9.2% in Brazil [18], 24% in Chile [19], and 24.9% in Republic of Korea [20] have been reported. Studies in the South American region have identified ESBLs such as CTX-M, TEM, SHV, PER-2, and AmpC in animals; the CTX-M-1, CTX-M-2, and CTX-M-8 enzymes are the most prevalent [13]. AmpCs are clinically important β-lactamases that hydrolyze first- and second-generation cephalosporins. These have been identified in intestinal *E. coli* isolates from small pets in which chromosomal mutations led to an enzyme overexpression responsible for AmpC resistance mechanisms [21,22]. Plasmid-mediated AmpC-producing (pAmpC) *E. coli* isolates have been identified in healthy dogs [16]. 

Carbapenemases, which are enzymes that hydrolyze carbapenems, have been identified in pet dogs; the main classes that have been described are: KPC, NDM, VIM, IMP, OXA-L48, and OXA-23 [23]. In companion animals as well as in humans, resistance to quinolones has been described more frequently in enterobacteria that are resistant to β-lactams. Multiple studies have shown the frequent association between genes that encode β-lactamases and several variants of genes that encode plasmid-mediated quinolone resistance (PMQR) such as the *qnr*A, *qnr*B, and *qnr*S genes [24,25].

Around 171 different sequence types (STs) have been reported in companion animals; ST38 and ST131 are found on all continents [26]. Various human-associated STs have been identified in domestic dogs [6,27,28,29,30]. Moreover, bacterial strains conduct intra- and inter-species genetic exchanges that include exchanges between dogs and their human owners [31,32,33]. Companion animals have been hypothesized as potential reservoirs of AMR bacteria, but there is a paucity of hard data. Studies conducted in Latin America showed high AMR to multiple antibiotics in intestinal strains of *E. coli* from small pets [34,35].

The purpose of the present study was the characterization of *E. coli* strains isolated from fecal samples of domestic dogs from the central region of Panama to investigate the AMR phenotype and molecular typing using the multilocus sequence typing (MLST) technique.

## 2. Materials and Methods

### 2.1. Strain Isolation

A cross-sectional study was conducted in August 2022. Fecal samples were obtained via rectal swabbing of domestic dogs from the central provinces of Herrera and Los Santos in the Republic of Panama with the informed consent of their owners. Samples were collected and transported in Cary-Blair medium (COPAN Diagnostics Inc.; Murrieta, CA, USA). Each sample was cultured on MacConkey agar (Merck Millipore; Darmstadt, Germany) and Chromocult^®^ (Merck Millipore; Darmstadt, Germany). Biochemical tests were conducted to identify *E. coli* strains using triple sugar iron agar, Simmons citrate agar, motility indole sulfide, and urea medium. For each sample collected, a technical sheet was completed anonymously with the following variables: age, sex, breed, and history of prior antibiotic treatment.

### 2.2. Antibiotic Susceptibility Test

The susceptibility to antibiotics was determined via the disk diffusion method on Muller Hinton agar (HiMedia Labs; Mumbai, India) [36], and the results obtained were interpreted according to the Clinical and Laboratory Standards Institute [37]. A total of 16 antibiotic discs were used: ampicillin (AMP, 10 µg), amoxicillin–clavulanate (AMC, 20/10 µg), aztreonam (ATM, 30 μg), cefepime (FEP, 30 µg), cefotaxime (CTX, 30 µg), cefoxitin (FOX, 30 µg), ceftazidime (CAZ, 30 μg), chloramphenicol (CPL, 30 µg), ciprofloxacin (CIP, 5 μg), gentamicin (GEN, 10 μg), imipenem (IPM, 10 μg), kanamycin (KAN, 30 μg), nalidixic acid (NAL, 30 µg), streptomycin (STS, 300 μg), tetracycline (TET, 30 μg), and trimethoprim–sulfamethoxazole (SXT, 1.25/23.75 µg). The sensitivity profiles allowed classification of the isolates as resistant, intermediate, and sensitive.

### 2.3. Molecular Typing Analyses and Molecular Identification of blaESBL/AmpC

The molecular typing analyses were performed using Pasteur’s MLST scheme. The MLST scheme was performed using a standardized protocol specific for *E. coli* [38]. The internal fragments of eight housekeeping genes (*din*B, *icd*A, *pab*B, *pol*B, *put*P, *trp*A, *trp*B, and *uid*A) were amplified from the chromosomal DNA of the *E. coli* strain. Sequencing of the polymerase chain reaction (PCR) products was performed with the services of Macrogen Inc. (Seoul, Republic of Korea). The gene sequences were analyzed using Geneious Prime v. 2020.5 (Biomatters, Ltd.; Auckland, New Zealand), and the allelic profiling was conducted using the Pasteur Institute’s *E. coli* Bacterial Isolate Genome Sequence Database (BIGSDB; https://bigsdb.pasteur.fr/ecoli/ecoli.html, accessed on 24 October 2022) for the specific MLST allele profiles and STs [39].

All isolates with β-lactam and quinolone resistance phenotypes were tested for *bla*CTX-M, *bla*TEM, *bla*SHV, *bla*CMY, AmpC, *qnr*A, *qnr*B, and *qnr*S [40,41,42] using PCR-specific primers (Table 1). The amplified samples were sequenced and analyzed using the BLASTN program of the National Center for Biotechnology Information (NCBI; https://blast.ncbi.nlm.nih.gov/Blast.cgi, accessed on 24 October 2022). For the *Amp*C gene, *E. coli* K-12 was used as the reference [21].

### 2.4. Statistical Analyses

The data were recorded in MS Excel (The Microsoft Corporation; Redmond, WA, USA). The data analyses were performed in Stata v. 11.0 (StataCorp, LLC; College Station, TX, USA). The descriptive statistics were calculated; the goodness-of-fit chi-square test was used to compare proportions with the alpha set at 0.05 for statistical significance.

## 3. Results

A total of 40 strains of *E. coli* were isolated from domestic dogs in central Panama. The mean age was 3.9 years (SD = 3.1) and most (*n* = 23; 58%) of the dogs were female. Most (*n* = 30; 75%) of the dogs had a known breed: Schnauzer (*n* = 7), Cocker (*n* = 4), Pitbull, Maltese, and Labrador (*n* = 3 each); 25% of the dogs had a mixed or unidentified breed. Almost half of the dogs (*n* = 17; 43%) had a history of prior antibiotic use; the most commonly used antibiotics reported were doxycycline (35%), cephalexin (20%), and enrofloxacin (10%).

Most (*n* = 32; 80%) of the *E. coli* strains analyzed showed resistance to at least one of the antibiotics analyzed; 20% (8/40) were sensitive to all antibiotics (*p* < 0.001). Figure 1 shows the proportion of sensitivity and resistance to the antibiotics analyzed in the isolated *E. coli* strains. We observed that 27.5% of the *E. coli* strains presented resistance to a single type of antibiotic, 15% to two types, 12.5% to three, 7.5% to four, 7.5% to five, 2.5% to six, and 7.5% to seven antibiotics. Table 2 shows the antimicrobial resistance phenotype of the *E. coli* strains analyzed; it can be seen that 2.5% (1/40) showed an ESBL phenotype, 40% (16/40) presented resistance to a β-lactam, and 20% (8/40) presented resistance to ciprofloxacin. The antibiotics with the highest prevalence of resistance were tetracycline (45%) and ampicillin (30%). Eight percent of the strains analyzed showed an extended-spectrum phenotype: one ESBL phenotype (LS09) and two extended-spectrum cephalosporins (ESCs) with an AmpC phenotype (HE04 and HE12 strains; Table 2). The HE02, HE04, and LS07 strains recorded resistance to imipenem; this will be analyzed in subsequent studies to elucidate possible resistance mechanisms not related to enzymes. No significant statistical differences were found between AMRs according to the dog breeds (*p* = 0.28).

Molecular typing using Pasteur’s MLST technique identified 14 STs in the *E. coli* samples analyzed (see Table 2). We most frequently observed ST399 (12%, 3/26), ST425 (12%, 3/26), ST910 (12%, 3/26), and ST960 (8%, 2/26). ST3 and ST88, which have been previously identified in human clinical isolates, also were found. We identified three new STs for the first time in this study: ST1015 (8%, 2/26), ST1016 (8%, 2/26), and ST1017 (8%, 2/26). Sequences of each of the three new STs of *E. coli* identified in this study can be found at https://bigsdb.pasteur.fr/cgi-bin/bigsdb/bigsdb.pl?page=profileInfo&db=pubmlst_ecoli_seqdef&scheme_id=1&profile_id=1015, (accessed on 24 October 2022); https://bigsdb.pasteur.fr/cgi-bin/bigsdb/bigsdb.pl?page=profileInfo&db=pubmlst_ecoli_seqdef&scheme_id=1&profile_id=1016, (accessed on 24 October 2022); and https://bigsdb.pasteur.fr/cgi-bin/bigsdb/bigsdb.pl?page=profileInfo&db=pubmlst_ecoli_seqdef&scheme_id=1&profile_id=1017, (accessed on 24 October 2022).

Table 3 shows the genotypes and phenotypes of the strains that showed resistance to β-lactams and quinolones. The *bla*TEM-1 was identified in 30% (3/10) of the strains with resistance to β-lactams. In addition, 80% (8/10) had a chromosomal AmpC with a mutation in the promoter regions −35 (1/10), −28 (1/10), −18 (5/10), −1 (5/10), and +58 (7/10), thereby registering different types of combinations: type 1 (−35, −18, −1, and +58), type 2 (−18, −1, and +58), type 3 (−28), and type 4 (+58); along with changes in the following bases: −35 (A → T), −28 (G → A), −18 (G → A), −1 (T → C), and +58 (T → C).

Among the strains with resistance to quinolones that were analyzed, we identified *qnr*S in two (20%) and *qnr*B in two (20%). The HE01 isolate showed the presence of both genes (*qnr*S and *qnr*B). Two of the three new STs identified in this study were carriers of antibiotic resistance genes: ST1016 (carrier of *bla*TEM-1) and ST1017 (carrier of *bla*TEM-1, *qnr*S, and *qnr*B).

## 4. Discussion

Antibiotics such as β-lactams and fluoroquinolones are widely prescribed to treat infections caused by *E. coli*. However, resistance to these groups of antibiotics is increasing in strains isolated from humans, animals, and the environment [15]. The present study showed bacterial phenotypes in combination with genetic characteristics of *E. coli* strains isolated from mostly healthy domestic dogs from the central region of Panama. We observed that a high percentage of the *E. coli* strains showed resistance to at least one of the antibiotics analyzed. Resistance to tetracycline and to ampicillin were the most prevalent, which was in agreement with a previous study in Latin America [35]. Additionally, resistance to ciprofloxacin and to cefotaxime were also identified. We found a prevalence of 2.5% for the ESBL phenotype for central Panama, which was one of the lowest in Latin America when compared to previous studies [16,18,19]. Resistance to these antibiotics has been previously described for *E. coli* isolated from healthy dogs in various regions of the world [35,43] as well as from environmental strains [5]. This aspect varies by country and depends greatly on the number and enforcement of public policies that regulate the prescription of antibiotics in veterinary medicine, which has been found to be related to the emergence of multidrug-resistant (MDR) strains [44]. Other factors that contribute to MDR in veterinary medicine include the use of antibiotics in animal feed, the use of antibiotics without quantification or empirical dosage, and the preference in the use of specific antibiotics such as doxycycline and enrofloxacin without alternating them with others [42,45]. Globally, MDR *E. coli* poses a challenge to health systems [46]. This study evidenced that a high proportion of dogs had a history of previous use of antibiotics; the most used were doxycycline, cephalexin, and enrofloxacin (a fluroquinolone). This study confirmed the presence of *E. coli* strains with an extended spectrum of resistance to β-lactam antibiotics (ESBL and AmpC phenotypes) in companion animals. Few studies have examined intestinal strains of *E. coli* in dogs under natural conditions. One study detected ESBL-producing *E. coli* from a single fecal sample from a healthy dog [47], while another study reported a prevalence of 9% of ESBL-producing *E. coli* strains in healthy dogs [48]. These findings are concerning due to the potential for transmission of antibiotic resistance determinants from commensal bacteria to potential pathogenic bacteria, zoonotic transmission, and opportunistic infections [49].

The MLST genetic sequences analyzed in this study identified 14 STs in *E. coli* strains, of which ST3 and ST88 have been previously reported in clinical isolates from hospitalized patients [50,51], which suggested that there was a genetic exchange of bacterial strains between human owners and their pets [7]. ST535, ST1017, ST1016, and ST88 showed the expression of plasmid resistance genes (β-lactamases and PMQR); ST1016 and ST1017 were identified for the first time in this study, which suggested that the population genetic diversity of bacterial clones favors antibiotic resistance through horizontal plasmid-transmission mechanisms. Additionally, our study showed that the PMQR variants (*qnr*B and *qnr*S) were coexpressed with a β-lactamase. It is noteworthy that the two strains identified as ST1017 presented coexpression of plasmid resistance genes (TEM/*qnr*S and TEM/*qnr*B), as has been found elsewhere [43,52,53,54,55]. As described previously, owners of companion animals and their family members may be at risk of acquiring bacteria with antibiotic resistance genes from their pets, especially if they are carriers of conjugative plasmids. A recent study showed that ESBL and *Amp*C plasmid genes acquired in the community were attributed in 7.9% of the cases to companion animals, especially dogs, which constitute the most common non-human source [12].

This study reported mutations of the *Amp*C gene in the promoter regions −35, −28, −18, −1, and +58 in 20% of the *E. coli* strains. Previous studies showed that these mutations can confer resistance to antibiotic groups such as penicillins, cephalosporins, cephamycins, and aztreonam [21,22,55]. Mutations were more prevalent among the new alleles described in this study and showed Type 2 as the most frequent pattern; this represented adaptive changes of *E. coli* at the environmental level due to antibiotic pressure and genetic exchange, which could contribute to the emergence of resistant strains.

The WHO established quinolones, β-lactams (such as third-, fourth-, and fifth-generation cephalosporins and amino-penicillins with and without β-lactamase inhibitors), and aminoglycosides—among others also used in veterinary medicine—as critically important antibiotics [3]. Our data showed resistance to several of the critically important antibiotics tested. This finding drew attention to potential difficulties in managing infections, as well as the importance of knowing the composition and distribution of antibiotic-resistance genotypes, as important factors in limiting the impact of AMR infections with *E. coli* in Panama. To the best of our knowledge, this was the first study in Central America on patterns of antimicrobial sensitivity and resistance as well as the molecular epidemiology of *E. coli* strains isolated from fecal samples of domestic dogs. The presence of genes from plasmids and chromosomal mutations for resistance to antibiotics, the identification of isolates previously reported in humans, and the genetic diversity of STs (including three new ones identified in this study) all revealed the determinants of resistance to antibiotics within the WHO’s *One Health* framework.

## Figures and Tables

**Figure 1 genes-14-00073-f001:**
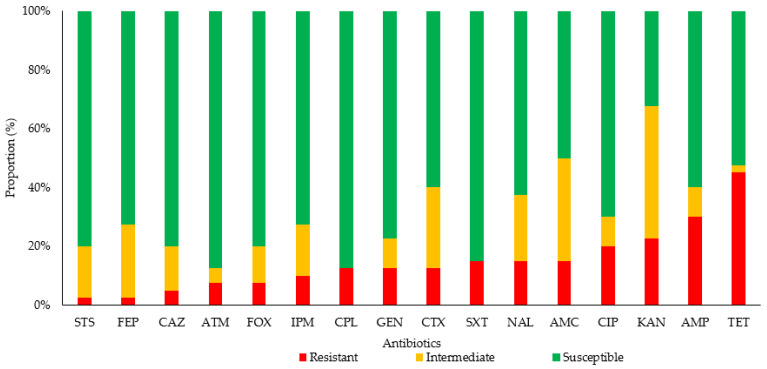
Proportion of antibiotic sensitivity and resistance in *E. coli* strains isolated from companion dogs. Abbreviations: AMC, amoxicillin–clavulanate; AMP, ampicillin; ATM, aztreonam; CAZ, ceftazidime; CIP, ciprofloxacin; CPL, chloramphenicol; CTX, cefotaxime; FEP, cefepime; FOX, cefoxitin; GEN, gentamicin; IPM, imipenem; KAN, kanamycin; NAL, nalidixic acid; STS, streptomycin; SXT, trimethoprim–sulfamethoxazole; TET, tetracycline.

**Table 1 genes-14-00073-t001:** β-lactamase genes analyzed.

Target	Primer Sequence (5′–3′)	Annealing Temperature (°C)	Product Size (bps)	Reference
CTX-M-F CTX-M-R	ATGTGCAGYACCAGTAARGTKATGGC TGGGTRAARTARGTSACCAGAAYSAGCGG	55	592	[40]
TEM-F TEM-R	GCGGAACCCCTATTTG ACCAATGCTTAATCAGTGAG	55	964	[41]
SHV-F SHV-R	TTATCTCCCTGTTAGCCACC GATTTGCTGATTTCGCTCGG	55	796	[42]
CMY-F CMY-R	ATGATGAAAAATCGTTATGCTGC GCTTTTCAAGAATGCGCCAGG	58	1138	[42]
AmpC1-71-F AmpC2120-R	AATGGGTTTTCTACGGTCTG GGGCAGCAAATGTGGAGCAA	55	191	[42]
*qnr*A-F *qnr*A-R	ATTTCTCACGCCAGGATTTG GATCGGCAAAGGTTAGGTCA	55	516	[42]
*qnr*B-F *qnr*B-R	GATCGTGAAAGCCAGAAAGG ACGATGCCTGGTAGTTGTCC	55	469	[42]
*qnr*S-F *qnr*S-R	ACGACATTCGTCAACTGCAA TAAATTGGCACCCTGTAGGC	55	417	[42]

Abbreviations: F, forward; R, reverse; bps, base pairs.

**Table 2 genes-14-00073-t002:** Phenotypes and genotypes of *E. coli* strains isolated from domestic dogs.

Isolate	Phenotypic Resistance Profile	Prior Antibiotics	Sequence Typing
HE01	AMP, CIP, CPL, IPM, SXT, TET	CEX, DOX, SFZ	535
HE02	AMP, CTX, IPM, KAN, TET	None	88
HE03	KAN, SXT	None	996
HE04	AMC, AMP, CIP, CPL, FOX, IPM, GEN	CEX, DOX, ENR	399
HE05	CPL, CTX, IPM, KAN	None	NA
HE06	—	DOX	129
HE07	—	DOX	NA
HE08	AMC, CPL	CEX, DOX, ENR	NA
HE09	AMP, SXT, TET	SPM	3
HE10	TET	DOX	473
HE11	TET	CEX, DOX	NA
HE12	AMC, AMP, ATM, CAZ, CTX, FOX	CEX, DOX	829
HE13	KAN, SXT, TET	None	910
HE14	TET	CEX, DOX, ENR	NA
HE15	—	CEX	NA
HE16	AMP	CEX	960
HE17	CIP, NAL, TET	None	526
HE18	AMC, FOX	None	425
HE19	—	None	NA
HE20	TET	None	NA
LS01	ATM, GEN, TET	ENR	910
LS02	CPL	None	399
LS03	NAL, TET	None	425
LS04	NAL, STS	DOX	399
LS05	TET	None	425
LS06	TET	None	NA
LS07	AMC, AMP, CIP, GEN, KAN, NAL, SXT, TET	DOX	1017 ^a^
LS08	—	DOX	1016 ^a^
LS09	AMC, AMP, ATM, CAZ, CTX, FEP	None	1016 ^a^
LS10	AMP, CIP, TET	None	NA
LS11	GEN	None	960
LS12	—	DOX	NA
LS13	KAN	DOX	NA
LS14	TET	None	910
LS15	—	None	NA
LS16	CIP, KAN, NAL	None	1015 ^a^
LS17	AMP, CIP, CTX, GEN, KAN	None	1015 ^a^
LS18	AMP, CIP, NAL, TET	None	1017 ^a^
LS19	—	None	NA
LS20	AMP, KAN, SXT, TET	None	NA

^a^ New alleles. Abbreviations: AMC, amoxicillin–clavulanate; AMP, ampicillin; ATM, aztreonam; CAZ, ceftazidime; CEX, cephalexin; CIP, ciprofloxacin; CPL, chloramphenicol; CTX, cefotaxime; DOX, doxycycline; ENR, enrofloxacin; FEP, cefepime; FOX, cefoxitin; GEN, gentamicin; IPM, imipenem; KAN, kanamycin; MLST, multilocus sequence typing; NA, not applicable (MLST was not conducted); NAL, nalidixic acid; SFZ, sulfadiazine; SPM, spiramycin; STS, streptomycin; SXT, trimethoprim–sulfamethoxazole; TET, tetracycline.

**Table 3 genes-14-00073-t003:** Resistance genes identified in the *E. coli* strains that demonstrated β-lactam and quinolone resistance.

Isolate	Sequence Typing	Mutations in the AmpC Promoter	β-Lactamases and PMQR	β-Lactam Resistance Pattern	Non-β-Lactam Resistance Pattern
HE01	535	—	TEM, *qnr*B, *qnr*S	AMP, IPM	CIP, CPL, NAL, SXT, TET
HE02	88	−35, −18, −1, +58 (Type 1) ^a^	TEM	AMP, CTX, IMP	KAN, TET
HE04	399	−18, −1, +58 (Type 2) ^a^	—	AMC, AMP, FOX, IMP	CIP, CPL, GEN
HE05	NA	—	—	CTX, IPM	CPL, KAN
HE12	829	−28 (Type 3) ^a^	—	AMC, AMP, ATM, CAZ, CTX, FOX	—
HE16	960	—	—	AMP	—
HE17	526	—	—	—	CIP, NAL, TET
HE18	425	+58 (Type 4) ^a^	—	AMC, FOX	—
LS02	399	—	—	—	CPL
LS03	425	—	—	—	NAL, TET
LS04	399	—	—	—	NAL, STS
LS07	1017 ^b^	−18, −1, +58 (Type 2) ^a^	TEM, *qnr*S	AMC, AMP	CIP, GEN, KAN, NAL, SXT, TET
LS09	1016 ^b^	+58 (Type 4) ^a^	TEM	AMC, AMP, ATM, CAZ, CTX, FEP	—
LS16	1015 ^b^	—	—	—	CIP, KAN, NAL
LS17	1015 ^b^	−18, −1, +58 (Type 2) ^a^	—	AMP, CTX	CIP, GEN, KAN
LS18	1017 ^b^	−18, −1, +58 (Type 2) ^a^	TEM, *qnr*B	AMP	CIP, NAL, TET

^a^ Specific promoter type; position numbered as defined by Muley et al. [22]. ^b^ New alleles. Abbreviations: AMC, amoxicillin–clavulanate; AMP, ampicillin; ATM, aztreonam; CAZ, ceftazidime; CIP, ciprofloxacin; CPL, chloramphenicol; CTX, cefotaxime; FEP, cefepime; FOX, cefoxitin; GEN, gentamicin; IPM, imipenem; KAN, kanamycin; MLST, multilocus sequence typing; NA, not applicable (MLST was not conducted); NAL, nalidixic acid; PMQR, plasmid-mediated quinolone resistance; STS, streptomycin; SXT, trimethoprim–sulfamethoxazole; TET, tetracycline.

## Data Availability

Not applicable.

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
