# Peer review of "Genetic Diversity and New Sequence Types of Escherichia coli Coharboring β-Lactamases and PMQR Genes Isolated from Domestic Dogs in Central Panama"

_genes, 2022, doi:10.3390/genes14010073_

Round 1

Reviewer 1 Report

I think the manuscript has valuable scientific value, however, I have some comments:

  1-The authors did not refer to the type of statistical test (they should mention the Chi-square test) also no paragraph on statistical analysis is included in the text.

2- The arrangement of the paragraphs is not correct as the materials and methods should be after the introduction.

3-There are several grammar mistakes

 4-The manuscript needs to add the gene sequences(Phylogenetic tree).

Also I have some commentes on the text

Author Response

REVIEWER 1

I think the manuscript has valuable scientific value, however, I have some comments:

  1-The authors did not refer to the type of statistical test (they should mention the Chi-square test) also no paragraph on statistical analysis is included in the text.

Response: Thanks for your valuable comments. We have added a paragraph on statistical analysis in the materials and methods section.

2- The arrangement of the paragraphs is not correct as the materials and methods should be after the introduction.

Response: Thanks for your valuable comments. We have now placed the “Materials and Methods” section after the “Introduction” and before the “Results” section.

3-There are several grammar mistakes

Response: Thanks for your valuable comments. Grammatical mistakes have been corrected.

 4-The manuscript needs to add the gene sequences(Phylogenetic tree).

Response: Thanks for your valuable comments. In the results section, links have been added to access the public database where the sequences of each new ST of E. coli identified in this study can be consulted at: https://bigsdb.pasteur.fr/cgi-bin/bigsdb/bigsdb.pl?page=profileInfo&db=pubmlst_ecoli_seqdef&scheme_id=1&profile_id=1015  ; https://bigsdb.pasteur.fr/cgi-bin/bigsdb/bigsdb.pl?page=profileInfo&db=pubmlst_ecoli_seqdef&scheme_id=1&profile_id=1016 ; https://bigsdb.pasteur.fr/cgi-bin/bigsdb/bigsdb.pl?page=profileInfo&db=pubmlst_ecoli_seqdef&scheme_id=1&profile_id=1017

Reviewer 2 Report

Manuscript title: New Escherichia coli Sequence Types isolated from domestic dogs in Panama

Manuscript ID: genes-2060435

Authors: Núñez-Samudio et al.

The study authored by Núñez-Samudio et al. describes isolation of E. coli from pet dogs and subsequent characterization of these isolates (AST, molecular detection of AMR genes and MLST). The is interesting and is of relevance considering the ever increasing threats posed by AMR worldwide. One important aspect the study addressed is the assignment of the STs to the isoltes which help in future epidemiological inference. Further, authors also reported mutations in the promoter regions of the AmpC gene which is also of relevance as mutations in promoter regions may lead to hyper-production of particular enzyme, thus enhancing the hydrolysis of the substrate by the enzyme.

Despite all the strengths, the manuscript suffers from several shortcoming too which are listed below.

If addressed by authors, the manuscript would be more suitable for the journal with international readership.

Title

1.      Title may be changed to emphasize new sequence types, not new Escherichia coli. Further, antimicrobial resistant nature of the isolated E. coli may also be highlighted

Abstract

1.      Typographical errors such as italicization, capitalization of first words of the sentences may be undertaken.

2.      Sentences in methods section of Abstract require restructuring e.g. " descriptive study that evaluated E. coli strains isolated from fecal samples of domestic dogs in central Panama"

3.      Sentences do not start with numerals - please correct e.g. "40% of the strains were resistant to 3 or more antibiotics. The most common resistance was to tetracycline (45%) and ampicillin (30%)."

4.      English language needs to be significantly improved to be suitable for international readership.

Key words

1.      'molecular genetic epidemiology' does not fit with the study, hence may be removed.

Introduction

1.      Introduction section lacks cohesiveness.

2.      The stated goal of the study was to "the characterization of E. coli strains isolated from fecal samples of domestic dogs in the central region of Panama, with the aim of investigating the AMR phenotype through identification of their susceptibility profile and characterization of the ESBL-producing strains with their molecular typing using the multi-locus sequence typing (MLST) technique". Yet, the authors dedicate one paragraph on WHO classification of antibiotics, another paragraph on implementation of One Health approach. These paragraphs may be substituted with information relevant to the study under report.

3.      While describing the status of ESBL producing enterobacteria, authors cite references from New Zealand and South Korea. However, references from nearby countries and from countries with similar socio-economic conditions would have been more appropriate.

4.      Authors also need to highlight the role of companion animals in dissemination of AMR.

5.      The paragraph stating the goals of the study need to be reworded for clarity.

Methods

1.      It was not clear, whether the authors submitted their sequence data to any public database. If they had submitted the sequences, the accession numbers may be provided. In case authors did not submit the sequences to any public database, the sequence files may be provided as supplementary data. This information would be crucial, as authors are reporting new STs and many mutations in promoter regions of AmpC gene.

Results

1.      In Figure 1. the numerical values on the vertical bars may be removed.

2.      In Table 2. under Sequence Typing column, authors reported many 'NAs'. It was not clear, whether authors performed MLST for these isolates. If authors performed MLST and did not get a match with the known STs, then it could be that these isolates were not E. coli. Authors are requested to check their data.

Discussion

1.      One Health was repeatedly highlighted in the study though that was not the stated goal. Authors also need to discuss the observed AMR patterns in relation to prior antibiotic use in the dogs.

2.      Was there any breed - AMR association? Authors may look into this also.

Author Response

REVIEWER 2

The study authored by Núñez-Samudio et al. describes isolation of E. coli from pet dogs and subsequent characterization of these isolates (AST, molecular detection of AMR genes and MLST). The is interesting and is of relevance considering the ever increasing threats posed by AMR worldwide. One important aspect the study addressed is the assignment of the STs to the isoltes which help in future epidemiological inference. Further, authors also reported mutations in the promoter regions of the AmpC gene which is also of relevance as mutations in promoter regions may lead to hyper-production of particular enzyme, thus enhancing the hydrolysis of the substrate by the enzyme.

Despite all the strengths, the manuscript suffers from several shortcoming too which are listed below.

If addressed by authors, the manuscript would be more suitable for the journal with international readership.

Title

  1. Title may be changed to emphasize new sequence types, not new Escherichia coli. Further, antimicrobial resistant nature of the isolated E. coli may also be highlighted

Response: Many thanks to the reviewer for valuable comments. We have now changed the title, emphasizing the new sequence types (ST) of E. coli and antimicrobial resistance genes found.

Abstract

  1. Typographical errors such as italicization, capitalization of first words of the sentences may be undertaken.

Response: Thank you for this suggestion. We have now rewritten all bacterial species names in italics and capitalized the first words of the sentences that required it.

  1. Sentences in methods section of Abstract require restructuring e.g. " descriptive study that evaluated E. coli strains isolated from fecal samples of domestic dogs in central Panama"

Response: Thank you for this suggestion. We have restructured the phrase to: “We report on a cross-sectional study that evaluated E. coli strains isolated from fecal samples of domestic dogs in central Panama”.

  1. Sentences do not start with numerals - please correct e.g. "40% of the strains were resistant to 3 or more antibiotics. The most common resistance was to tetracycline (45%) and ampicillin (30%)."

Response: Thanks for your valuable comment. We have now changed the sentence that start with numeral “40%” to “Forty percent”

  1. English language needs to be significantly improved to be suitable for international readership.

Response: Thanks for your valuable comment. We have now completely revised and corrected the English of the manuscript.

Key words

  1. 'molecular genetic epidemiology' does not fit with the study, hence may be removed.

Response: Thanks for your valuable comment. In the keywords section, we have now removed “molecular genetic epidemiology” and added “multilocus sequence typing” and “Escherichia coli”.

Introduction

  1. Introduction section lacks cohesiveness.

Response: Thanks for your valuable comment. We have improved the introduction by making it more cohesive.

  1. The stated goal of the study was to "the characterization of E. coli strains isolated from fecal samples of domestic dogs in the central region of Panama, with the aim of investigating the AMR phenotype through identification of their susceptibility profile and characterization of the ESBL-producing strains with their molecular typing using the multi-locus sequence typing (MLST) technique". Yet, the authors dedicate one paragraph on WHO classification of antibiotics, another paragraph on implementation of One Health approach. These paragraphs may be substituted with information relevant to the study under report.

Response: Thanks for your valuable comment. As suggested by the reviewer, we have removed the paragraphs referring to WHO and "One Health" and replaced it with an introduction with paragraphs referring to antibiotic resistance in E. coli strains isolated from companion animals in Latin America and worldwide.

  1. While describing the status of ESBL producing enterobacteria, authors cite references from New Zealand and South Korea. However, references from nearby countries and from countries with similar socio-economic conditions would have been more appropriate.

Response: Thanks for your valuable comment. We have now included references on the prevalence of ESBL strains of E. coli from Latin American countries such as Brazil, Chile and Mexico.

  1. Authors also need to highlight the role of companion animals in dissemination of AMR.

Response: Thanks for your valuable comment. As suggested by the reviewer, we have given greater prominence to the role of companion animals in the dissemination of AMR strains of E. coli.

  1. The paragraph stating the goals of the study need to be reworded for clarity.

Response: Thanks for your valuable comment. For the sake of clarity, we have reworded the paragraph setting out the goals.

Methods

  1. It was not clear, whether the authors submitted their sequence data to any public database. If they had submitted the sequences, the accession numbers may be provided. In case authors did not submit the sequences to any public database, the sequence files may be provided as supplementary data. This information would be crucial, as authors are reporting new STs and many mutations in promoter regions of AmpC gene.

Response: Thanks for your valuable comments. In the results section, links have been added to access the public database where the sequences of each new ST of E. coli identified in this study can be consulted at: https://bigsdb.pasteur.fr/cgi-bin/bigsdb/bigsdb.pl?page=profileInfo&db=pubmlst_ecoli_seqdef&scheme_id=1&profile_id=1015  ; https://bigsdb.pasteur.fr/cgi-bin/bigsdb/bigsdb.pl?page=profileInfo&db=pubmlst_ecoli_seqdef&scheme_id=1&profile_id=1016 ; https://bigsdb.pasteur.fr/cgi-bin/bigsdb/bigsdb.pl?page=profileInfo&db=pubmlst_ecoli_seqdef&scheme_id=1&profile_id=1017

Results

  1. In Figure 1. the numerical values on the vertical bars may be removed.

Response: Thanks for your valuable comment. We have now removed the numerical values on the vertical bars.

  1. In Table 2. under Sequence Typing column, authors reported many 'NAs'. It was not clear, whether authors performed MLST for these isolates. If authors performed MLST and did not get a match with the known STs, then it could be that these isolates were not E. coli. Authors are requested to check their data.

Response: Thanks for your valuable comment. MLST is expensive in our setting, so it was mainly performed on strains showing the main antibiotic resistance phenotype profiles, such as ESBL, ESC, quinolone resistance, AmpC and others. NA means that MLST was not conducted.

Discussion

  1. One Health was repeatedly highlighted in the study though that was not the stated goal. Authors also need to discuss the observed AMR patterns in relation to prior antibiotic use in the dogs.

Response: Thanks for your valuable comment. We have removed references to One Health throughout the article.

  1. Was there any breed - AMR association? Authors may look into this also.

Response: Thanks for your valuable comment. No significant statistical differences were found between AMRs according to breed (p 0.28).